# Pharmacogenomics in Psychiatry Practice: The Value and the Challenges

**DOI:** 10.3390/ijms232113485

**Published:** 2022-11-03

**Authors:** Aminah Alchakee, Munazza Ahmed, Leen Eldohaji, Hamid Alhaj, Maha Saber-Ayad

**Affiliations:** 1College of Medicine, University of Sharjah, Sharjah 27272, United Arab Emirates; 2Sharjah Institute for Medical Research, Sharjah 27272, United Arab Emirates; 3College of Pharmacy, University of Sharjah, Sharjah 27272, United Arab Emirates

**Keywords:** cytochrome P450, CYP2D6, CYP2C19, antidepressants, antipsychotics

## Abstract

The activity of cytochrome P450 enzymes is influenced by genetic and nongenetic factors; hence, the metabolism of exogenous psychotropic medications and potentially some endogenous neuropeptides is variably affected among different ethnic groups of psychiatric patients. The aim of this review is to highlight the most common cytochrome P450 isoenzymes associated with the metabolism of psychotropic medications (antidepressants, antipsychotics, and mood stabilizers), their variations among different populations, their impact on endogenous neurotransmitters (dopamine and serotonin), and the effect of nongenetic factors, particularly smoking, age, and pregnancy, on their metabolic activity. Furthermore, the adverse effects of psychiatric medications may be associated with certain human leukocytic antigen (HLA) genotypes. We also highlight the gene variants that may potentially increase susceptibility to obesity and metabolic syndrome, as the adverse effects of some psychiatry medications. Collectively, the literature revealed that variation of CYP450 activity is mostly investigated in relation to genetic polymorphism, and is directly correlated with individualized clinical outcomes; whereas adverse effects are associated with HLA variants, projecting the value of pharmacogenetics implementation in psychiatry clinics. Only a few previous studies have discussed the impact of such genetic variations on the metabolism of endogenous neuropeptides. In this review, we also report on the prevalence of key variants in different ethnicities, by demonstrating publicly available data from the 1000 Genomes Project and others. Finally, we highlight the future direction of further investigations to enhance the predictability of the individual gene variants to achieve precision therapies for psychiatric patients.

## 1. Introduction

Psychiatric disorders are prevalent and associated with high levels of morbidity and mortality. Conditions such as depression and anxiety are among the leading causes of disease burden worldwide [1,2]. According to statistics reports of the World Health Organization (WHO), more than 264 million patients suffer from depression [3]. The Global Burden of Diseases measures the burden of disorders by using a disability-adjusted life-year (DALY) metric, which quantifies the burden of a disease in terms of mortality and morbidity [4,5]. The highest DALY was reported for major depressive disorders in the age group of 30–34 years in 2020 [6]. Increased burden of mental disorders has been recognized to be globally persistent since 1990 [4].

In recent years, significant discoveries have been made in the management of psychiatric conditions, including psychological, pharmacological, and physical treatments. Effective psychotropic drugs, such as antidepressants, mood stabilizers, and antipsychotics, are commonly used to treat several psychiatric disorders. However, one of the major challenges for the prescribing clinician is how to select a safe and effective treatment option tailored to the needs of each patient. Unfortunately, many patients often go through a trial-and-error process characterized by poorly controlled symptoms and/or severe drug responses before the most suitable psychotropic drug and doses are established [7].

Individualizing treatment plans for psychiatric patients by implementing pharmacogenomic (PGx) testing with the aim of prescribing precision therapies is the focus of the newly developed field of precision psychiatry [8,9]. This promising alternative to conventional psychiatric prescribing utilizes our understanding of how certain genes and specific biomarkers may influence the individual’s response to medications, paving the way to more individually tailored approaches. For example, evidence supports the use of single nucleotide polymorphisms (SNPs) to measure treatment response and potential adverse drug reactions for antidepressants. Furthermore, it has been proposed that multi-omics and neuroimaging data can be used as biomarkers to predict responses based on newly developed artificial intelligence and deep learning frameworks [10]. However, wide adoption of pharmacogene testing has not yet occurred in psychiatry, which may be due to a number of factors, including varying knowledge of genetics among psychiatrists, differing opinions on the efficacy of pharmacogenomic testing in clinical practice, and the presence of conflicting perceptions of the PGx tool evidence-base [5,9,11].

In this review, we critically discuss the recent advances in our understanding of pharmacogenomics in psychiatry. We focus on the activity of cytochrome P450 enzymes, how they may be influenced by genetic and non-genetic factors, their influence on the metabolism of psychotropic medications, and the variation among different populations. We will also highlight the impact of HLA gene variation in predicting the potential adverse effects of psychotropic medications. Finally, we will discuss genetic susceptibility to obesity and metabolic syndrome, since both are recognized adverse effects of several psychotropic medications, especially second generation anti-psychotics. 

### 1.1. CYP450 Genetic and Phenotypic Variations

Cytochrome P450 (CYP450) enzymes are hemoproteins found in different human tissues, including intestines, kidneys, plasma, lung, and mainly in the liver [12]. Their role encompasses detoxifying several endogenous and exogenous substances by oxidation, hydroxylation, epoxidation, and dealkylation mechanisms. Genetic variability of CYP enzymes highly interferes with enzymatic activity of different polymorphisms, resulting in personalized metabolism manner [13,14]. Allelic variants of CYP enzymes are commonly named according to the (*) system and translated to different phenotypes, including ultrarapid metabolizers (UMs), rapid metabolizers (RMs), normal metabolizers (NMs), intermediate metabolizers (IMs), and poor metabolizers (PMs) [8,15]. In fact, several mutations are caused by CYPs polymorphisms, such as alternative splicing and frame shifting, resulting in an altered structure and function of the enzymes [16]. In particular, structural modification of the binding site regions has been noticed in multiple polymorphisms of CYP2C19, CYP2D6, and CYP2C9, as shown in Figure 1 [17]. For instance, CYP2C19*5B (rs56337013) and CYP2C19*8 (rs41291556) alleles are known to be catalytically inactive due to mutations in the heme binding site (Figure 1A) [17,18]. Despite the fact that some alleles of CYP2D6 are associated with mutations in the binding sites (Figure 1B), recent studies of PGx in clinical psychiatry have focused on measuring the copy-number of CYP2D6, such as gene duplication and deletion, due to the complexity of studying SNPs of CYP2D6 locus [19,20]. Remarkably, more than 50,000 CYP enzymes are found in nature. Around fifty-seven CYP450 genes have been investigated in humans; six of them are mainly involved in drug biotransformation, including CYP 2C9, 2D6, 2C19, 3A4,1A2, and 2E1. The most abundant human CYP enzymes are CYP2D6 and CYP3A4 [13,20]. In particular, hepatic CYP isoenzymes commonly metabolize psychotropic medications: antipsychotics (CYP2D6), anticonvulsants (CYP2C9), and antidepressants (CYP2C19 and CYP2D6) [21]. Although most psychotropic medications are metabolized in the liver, some of them, such as lithium, are directly eliminated by the kidneys [22].

Moreover, advancements in pharmacogenomics research have progressively led to the corroboration of not only inter-individual but inter-ethnic differences in drug pharmacokinetics and pharmacodynamics. Considering the inter-ethnic differences in allele frequencies, the prevalence of certain genes in one ethnic group over the other can help prioritize those in need of testing or prescribing. This approach has been endorsed by the FDA [23] and clinical guidelines through the Clinical Pharmacogenetics Implementation Consortium (CPIC) [7] which recommend genetic screening before prescribing certain drugs, such as carbamazepine. Most studies focus on genetic variations of the Drug Metabolizing Enzymes (DMEs) CYP2C9, CYP2C19, CYP2D6, which are, primarily, responsible for phase I metabolism of around 40% of drugs in clinical use. Hence, several variant alleles have been uncovered; the most frequently reported are summarized in Table 1.

**Table 1 ijms-23-13485-t001:** Allele categories based on common star alleles for selected CYP enzymes (6). The data was obtained from pharmgkb.org (accessed on 5 February 2022). PharmGKB employs a framework of nine biogeographical groupings to annotate data on participants in pharmacogenomic research in terms of race and ethnicity. The study that guided the construction of the grouping system employed data from the Human Genome Diversity Project and the 1000 Genomes Project [24,25].

Star Allele Functionality		Diplotype Functionality	
CYP2D6			
Type of Allele	Examples of Alleles	Phenotype	Diplotype
Normal function	*1, *2	Normal metabolizer	*1/*1, *1/*2
Decreased function	*9, *10, *17, *41	Intermediate metabolizer	*1/*3, *9/*17
No function	*3, *4, *5, *6	Ultrarapid metabolizer	*1/*1 × 2
CYP2C9			
Type of Allele	Examples of Alleles	Phenotype	Diplotype
Normal function	*1	Normal metabolizer	*1/*1
Decreased function	*2, *5, *8, *11	Intermediate metabolizer	*1/*2, *1/*11
No function	*3, *6, *13	Poor metabolizer	*2/*3, *3/*5
CYP2C19			
Type of Allele	Examples of Alleles	Phenotype	Diplotype
Normal function	*1	Normal metabolizer	*1/*1
Increased function	*17	Intermediate metabolizer	*1/*2, *1/*3
No function	*2, *3	Rapid metabolizer	*17/*17
CYP2B6			
Type of Allele	Examples of Alleles	Phenotype	Diplotype
Normal function	*1, *2, *5	Normal metabolizer	*1/*1, *1/*2
Increased function	*4	Intermediate metabolizer	*4/*8
No function	*8	Ultrarapid metabolizer	*4/*4

**Figure 1 ijms-23-13485-f001:**
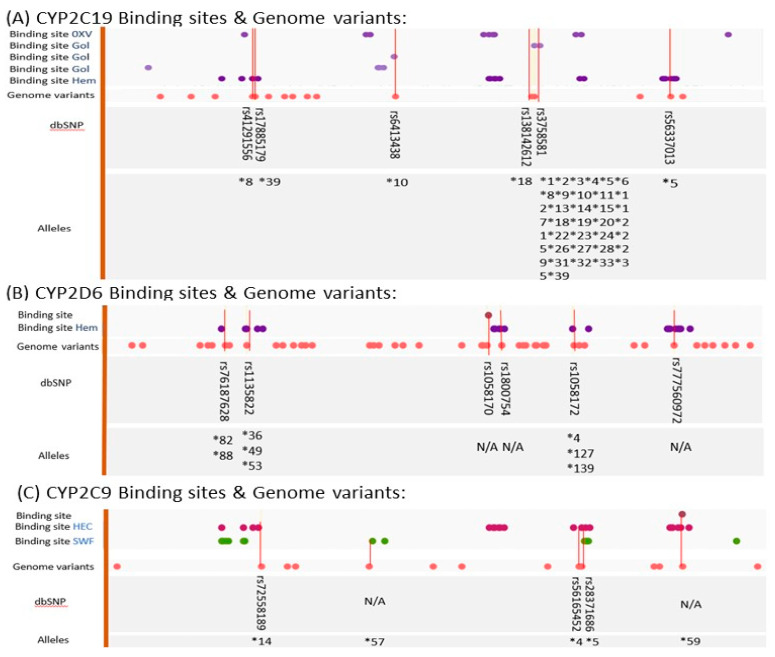
Illustration provides an insight into correlation between binding sites named by their unique ligands (OXV: (4-hydroxy-3,5-dimethylphenyl)(2-methyl-1-benzofuran-3-yl)methanone, GOL: glycerol, HEM: protoporphyrin ix containing Fe, HEC: heme c and SWF: s-warfarin) and genome variations (defined as dbSNP number) deposited in PDB (retrieved from Protein Data Bank) [17,18,19,20,26].

### 1.2. CYP450 Genetic Variations and Antidepressants

Selective serotonin reuptake inhibitors (SSRIs) and serotonin and noradrenaline reuptake inhibitors (SNRIs) are among the most prescribed antidepressant medications. At higher doses, they may be associated with mild to severe potential side effects, such as sweating, serotonin toxicity, and sexual dysfunction [27,28]. All SSRIs and SNRIs are metabolized in the liver by different CYP450 isoenzymes. SSRIs, citalopram, and escitalopram are mainly metabolized by the highly polymorphic cytochrome 2C19. Consequently, the clinical Pharmacogenetics Implementation Consortium (CPIC) published gene-based therapeutic guidelines for the SSRIs citalopram/escitalopram, based on the CYP2C19 genotype [28,29]. Genetic variations in the CYP2C19 gene may cause an increase or decrease in the metabolic activity of the CYP2C19 enzyme, and thereby, the drug plasma concentration [21]. For example, the wild-type allele CYP2C19*1 encodes a fully functional enzyme with normal phenotypic activity; whereas the *17 variation is associated with higher enzyme activity, and *2 with no activity. The phenotypes associated with star alleles are stated in Table 1. Therefore, depressed patients with poor metabolizing phenotypes demonstrate high citalopram plasma concentration. Thus, the FDA recommends a 50% dose reduction for poor metabolizers to avoid the risk of developing QT prolongation [21,30,31]. Furthermore, fluvoxamine is metabolized by CYP2D6. The CPIC dosing guidelines suggest a 25–50% reduction in the recommended initial dose of the fluvoxamine for poor metabolizers and titrate until maximum effective dose to avoid unwanted adverse effects, or to use an alternative medicine not metabolized by CYP2D6 [31,32]. 

Venlafaxine and paroxetine are also metabolized by the CYP2D6 enzyme (Table 2). Moreover, paroxetine is capable of inhibiting the CYP2D6 enzyme; when it is given in combination with the selective noradrenaline reuptake inhibitor, atomoxetine, higher steady state concentrations of atomoxetine have been observed [33,34]. 

Tricyclic antidepressants (TCAs), including, imipramine, amitriptyline, trimipramine, and clomipramine, are among the earliest approved antidepressant agents which are primarily metabolized by CYP2C19, as summarized in Table 2. However, the CYP2D6 pathway is essential to catalyze further metabolic pathways [35,36]. On the other hand, the relatively newer antidepressant drug, mirtazapine, is mainly metabolized by CYP2D6, CYP1A2, and CYP3A4 isoenzymes [37].

**Table 2 ijms-23-13485-t002:** Summary of the main metabolizing Cyp450 isoenzyme of psychotropic agents and endogenous neurotransmitters.

Main Enzyme	Medications and Transmitters Affected (Paired to the Gene)				
	Anti-Depressant	Anti-Psychotics	Mood-Stabilizer	Neurotransmitter	References
CYP2C19	Imipramine, Amitriptyline, Trimipramine, Clomipramine, Citalopram, Escitalopram	-	-	Dopamine and Serotonin	[28,29]
**CYP2D6**	Amitriptyline, Fluvoxamine, Paroxetine, Venlafaxine, Fluoxetine, Mirtazapine, Vortioxetine.	Aripiprazole, Risperidone, Brexpiprazole	-	Dopamine	[15,35,38]
**CYP1A2**	Amitriptyline, Clomipramine, Trimipramine, Imipramine, Doxepin, Mirtazapine.	Olanzapine, Asenapine, Clozapine	-	-	[39,40]
**CYP3A4**	Mirtazapine	Haloperidol, Clozapine, Aripiprazole, Quetiapine, Levomepromazine, Brexpiprazole	Carbamazepine	-	[39,41,42]
**CYP2C9**	-	-	Valproic acid	-	[43]
**CYP2C8**	-	-	Carbamazepine	-	[44]
**CYP2B6**	Sertraline	-	-	-	[45]

### 1.3. CYP450 Genetic Variations and Antipsychotics

Antipsychotics are a class of psychotropic medications used to treat multiple psychiatric conditions, including schizophrenia spectrum disorders, bipolar disorder, and major depression [46]. Psychiatric chemotherapy is strongly influenced by variations in CYP2D6 and CYP2C19 genes; thus showing individualized drug efficacy and safety profile [47]. Around 40% of antipsychotic medications are metabolized by the highly polymorphic CYP2D6 enzyme [48]. Accordingly, food and drug administration labeled 24 antipsychotic dose recommendations, including clozapine, to be prescribed based on variant individual phenotypes [49], in addition to reducing the haloperidol dose by half for CYP2D6 PM phenotypes [50,51]. Several haplotypes of CYP2D6, CYP1A2, CYP3A5, and CYP3A4 significantly influence antipsychotic drug metabolism [52]. In particular, CYP1A2 metabolizes mainly olanzapine, asenapine, and clozapine (Table 2). CYP2D6 has a major role in the metabolism of aripiprazole and risperidone, while CYP3A4 metabolizes mainly aripiprazole, clozapine, quetiapine, and levomepromazine [53]. In this regard, the missense mutation of the rs680055 variant of CYP3A4 highly impacts antipsychotic response [54]. Conversely, quetiapine and aripiprazole are the least affected drugs by SNP of CYP450 genes; however, a patient’s genotype should be taken into consideration to minimize antipsychotic drugs harm [53]. 

One study clinically assessed a psychiatric patient’s response before and after performing pharmacogenetic testing for CYP2C19 and CYP2D6; based on judgments made by physicians, 23% reported improvement in patient outcomes, and 41% reported no change in term of improvement. Importantly, none of them reported worse outcomes upon utilizing pharmacogenetic testing [48]. Another study, conducted in 868 patients diagnosed with depression taking either nortriptyline or escitalopram, found that CYP450 genotyping analysis was ineffective in predicting any adverse drug reaction [55]. In this context, CYP2D6 genotyping exhibited no reduction in the risk of hyperprolactinemia, which may be induced by many antipsychotics as a side effect [56]. Moreover, Walden et al. stated that utilizing pharmacogenomic testing to predict the occurrence of side effects was statically insignificant [48]. 

Most antipsychotic agents are given orally; therefore, analyzing pharmacogenetic parameters of antipsychotics must include studying different families of xenobiotic transporter genes instead of focusing only on CYP450 genes [57]. For instance, the mutation of HTR2C (serotonin receptor) is associated with weight gain and metabolic syndrome. In addition, high prolactin levels have been investigated in patients with DRD2*A1 allele (dopamine receptor); interestingly, patients on clozapine showed hyperprolactinemia, although it was least likely to increase prolactin levels [58,59].

### 1.4. CYP450 Genetic Variations and Mood Stabilizers

Valproic acid (VPA) and Carbamazepine (CBZ) are commonly prescribed mood stabilizers, especially for bipolar disorder, which is characterized by mood alternations between depression and mania. However, the response and development of side effects to CBZ and VPA are highly variable among patients due to drug metabolism heterogeneity among different populations [60,61]. For example, Japanese women with CYP2C19*3 or CYP2C19*2 alleles are more susceptible to gain weight during valproate therapy. These alleles significantly correlated with variant VPA plasma concentrations. Therefore, detection of the SNPs of CYP2C19 is likely to be beneficial to optimize the valproic blood concentration; hence, controlling drug responses and side effects [60]. In addition, CYP2B6 and CYP2A6 are responsible for producing only 20–25% of valproic acid metabolites; however, the CYP2C9*3 allelic variant is primarily associated with the formation of hepatotoxic 4-ene-VPA metabolite, which is more potent than the VPA substrate [60]. 

Although carbamazepine metabolism is catalyzed by CYP2C8 and CYP3A4, to form the active equipotent CBZ metabolite, EPHX1 (epoxide hydrolase) gene is considered the major CBZ metabolizer. Notably, the influence of CYP3A4 and EPHX1 polymorphisms is still under investigation [62,63]. Furthermore, individuals with CYP3A5 variation exhibited a wide range of CBZ responses; the non-functional CYP3A5*3 allele highly influences CBZ blood concentration, in addition to using CBZ in combination with enzyme inducers which showed a different CBZ clearance rate [63,64,65]. It has been suggested that individuals’ susceptibility to developing CBZ side effects could be measured by genotyping CYP2C19, CYP3A5, and EPHX1 [60]. 

Lithium, being a generally effective mood stabilizer, is quite unique since its renal metabolism bypasses the CYP450 system. While it is still recommended for use as a first line mood stabilizer [66], it is worth noting that its long-term use is associated with an impairment of thyroid and renal functions [52]. The findings of lithium-gene association studies revealed the importance of genetic factors in lithium response. As yet, the exact genes that co-segregate with lithium responses have not been fully detected [53]. 

## 2. The Influence of CYP2D6 and CYP2C19 Variants on Neurotransmitters

Generally, CYP450 enzymes contribute to cellular homeostasis by metabolizing several endogenous compounds, including, dopamine, serotonin, cortisol, progesterone, and testosterone [67]. Importantly, dopamine and serotonin are incapable of crossing the blood-brain barrier, suggesting the presence of CYP450 enzymes in the brain. Particularly, in the brain tissues, human CYP2D6 demonstrated the ability to catalyze the aromatic hydroxylation of tyramine to dopamine (DA), and O-demethylation of 5-methoxytryptamine to serotonin (5-HT) [68]. As a result, entire human behaviors, including personality and neuropsychiatric disorders such as schizophrenia (SCZ), major depression (MD), and obsessive-compulsive disorder (OCD), are influenced by the highly observed CYP2D6 variations [69]. In contrast, CYP2C19 is expressed in human fetal brain tissue and disappears after birth; therefore, CYP2C19 is involved in brain neurodevelopment and significantly influences adult depressive phenotypes. In particular, the absence of CYP2C19 is correlated with a lower prevalence of depression. For example, one of the most common CYP2C19 alleles among Swedish subjects is CYP2C19*2. It is characterized by an inactive CYP2C19 enzyme; thus, lowering susceptibility to depressive moods [70,71,72]. A recent study examined the genetic impact of CYP2D6 polymorphism on the susceptibility of individuals to develop schizophrenia, and suggested that CYP2D6 variations may alter the structure of the hippocampal white matter region of the brain and the neurotransmission of dopamine; thus, highlighting neuronal connectivity underlying the pathophysiology of schizophrenia [73]. For instance, PMs have a greater DA tone in the pituitary gland, combined with a lower serotonin tone, due to serotonin-mediated tonic inhibition [74]. Overall, the contributions of CYP2D6 and CYP2C19 in the metabolism of endogenous substances are not fully understood, and further investigations are required to prove the physiological implications of CYP450 in the brain [69,72,75]. 

### 2.1. Dopamine Synthesis via CYP2D6

Dopamine is a neurotransmitter that also functions as a precursor to noradrenaline, and adrenaline is synthesized initially from phenylalanine, which is converted by phenylalanine hydroxylase to tyrosine and then oxidized to dihydroxyphenylalanine (L-DOPA) by tyrosine hydroxylase; L-DOPA is ultimately metabolized by DOPA decarboxylase to dopamine [68]. Alternatively, dopamine can be formed in the brain from p- and m-tyramine through aromatic hydroxylation by CYP2D6 [75]. Notably, among CYP450, only CYP2D6 catalyzes the synthesis of dopamine [67]. 

### 2.2. Serotonin Metabolism via CYP2D6 and CYP2C19

Serotonin is a fundamental neurotransmitter that has been implicated in impulsive behavior, as well as in depressive and anxiety disorders. It is found in both vertebrate and invertebrate neural systems [76]. Serotonergic pathways in the brain are initiated from 5-HT, containing groups (B1–B9) of neurons of the raphe nuclei in the brain stem; subsequently, serotonin concentration mainly depends on free plasma tryptophan levels [77]. In vivo and in vitro studies have documented the mediation of CYP2D6 to regenerate serotonin from 5-methoxytryptamine (5-MT), suggesting the importance of CYP2D6 in neuropsychological events on the central nervous system (CNS), as well as drug metabolism among different individuals [68]. Notably, various CYP2D6 activities in human populations have been correlated to different personality traits; people with CYP2D6 Ems were less anxious and more socially successful than PMs [76]. Interestingly, in vivo investigations revealed that ultrarapid metabolizers (Ums) had higher serotonin levels in their platelets than extensive metabolizers (Ems) and poor metabolizers (PMs) [74]. It was also suggested that CYP2C19 is involved in the biotransformation of serotonin and is correlated with bilateral hippocampal volume. Increasing CYP2C19 expression is noticed with 5-HT1A downstream signaling and reduction of hippocampal volume. However, further detailed studies are required to confirm the role of CYP2C19 in 5-HT1A biochemical signaling, and consequently, hippocampal volume and depression [70]. The effect of gene variants on the metabolism of dopamine and serotonin (and potentially other neurotransmitters) adds to the complexity of predictions of the effect of such variants on the overall response of the patient to medications [71]. Nonetheless, following dosing recommendations based on an individual’s phenotype can aid in optimizing psychotherapy. Table 3 provides a list of dosing recommendations for psychotropic drugs based on CPIC guidelines for CYP2C19, CYP2D6, and CYP2C9 phenotypes. Table 4 lists the drugs that can be used as alternatives, so as to minimize the likelihood of pharmacokinetic variability accounted for by the CYP isoenzymes.

## 3. Human Leukocyte Antigen (HLA) Gene in Psychiatry

The human leukocyte antigens (HLA) are a group of genes encoding the major histocompatibility complex proteins (MHC), which play a crucial role in immune and inflammatory responses [87]. The HLA variants should be added to the list of pharmacogenomics panels to be tested in psychiatry patients, to ensure a safe and effective individualized therapy.

Recent studies have demonstrated the impact of genetic variations of the HLA gene cluster on the etiology of several psychiatric disorders. Interestingly, data showed that the highly variable HLA molecules play a major role in the etiology of bipolar disorder and schizophrenia, but not in depressive disorders and attention deficit hyperactivity disorder (ADHD) [88]. In particular, HLA molecules were found to modulate neural signaling and synaptic integration, thus affecting congenital abilities such as behavior, learning, and memory. However, the exact contribution is not yet fully understood [89,90].

Additionally, HLA genetic diversity is associated with psychotropic treatment response and developing adverse drug reactions (ADRs). For instance, Class I and II HLA alleles have been shown to partially mediate clozapine-induced agranulocytosis [91]. A recent study showed a correlation between double amino-acid variants at positions 62 and 66 of HLA-A peptide-binding groove and a better response to treatment with Risperidone in schizophrenia patients [92]. HLA polymorphisms among different ethnic groups is significantly associated with SCARs (Severe Cutaneous Adverse Drug Reactions), including Stevens-Johnson syndrome (SJS) and the toxic epidermal necrolysis (TEN), the life threating adverse drug reactions presenting as serious skin hypersensitivity reactions [93,94]. 

Therefore, the implementation of pre-emptive testing of HLA genotyping in clinical practice might prevent these side effects [95]. Recently, in neuropsychiatric medications, Phenytoin, Carbamazepine, and Oxcarbazepine, with the following biomarkers (HLA-B*15:02, HLA-A*31:01/HLA-B*15:02 and HLA-B*15:02/HLA-A*31:01), respectively, have been the most documented with SJS/TEN [94].

## 4. Gene Variants Predicting Predisposition to Obesity and Metabolic Syndrome

We expect more gene variants will be added soon to the pharmacogenomics panel for psychiatric patients; some of the most significant are those associated with obesity and metabolic syndrome. Research has found that patients with psychiatric disorders are known to have higher morbidity and mortality compared to the overall population [96]. This is likely due to the metabolic syndrome that these patients experience, predisposing them to cardiovascular diseases, type 2 diabetes, hypertension, dyslipidemia, hyperglycemia, and obesity [97]. Various aspects impact this high comorbidity, including genetic factors. Polymorphisms of different genes are known to be associated with the development of metabolic syndromes in psychiatric patients among different ethnic populations [98].

### 4.1. Genes Associated with Obesity

The fat mass and obesity-associated (FTO) gene encodes for the alpha-ketoglutarate-dependent dioxygenase. Monogenic disorders related to mutations in the FTO regions have also been identified in humans [99], and polymorphisms in the noncoding areas of this gene have been linked with obesity and several other diseases, particularly those for which obesity is a risk factor [100]. While FTO gene polymorphisms associated with obesity have substantial variability in allele frequencies among ethnoterritorial groups, comparable allele recording frequencies of multiple SNPs across representatives of the same ethnoterritorial group also exist. The diversity between subpopulations within territorial groupings is similarly negligible for most SNPs; independent of interterritorial variations in allele recording frequency. 

It is important to note that the FTO gene polymorphisms linked to characteristics are found in noncoding regions that have no effect on the structure or function of alpha-ketoglutarate-dependent dioxygenase. Nonetheless, BMI and other anthropometric features representing the degree of obesity have been reported to be linked to rs1421085. The T-to-C substitution results in a twofold increase in the expression of two genes distal to FTO, namely IRX3 and IRX5 [101]. During preadipocyte differentiation, the proteins encoded by the IRX3 and IRX5 genes shift their development from energy-dissipating to energy-storing adipocytes; an increase in IRX3 and IRX5 expression also leads to an increase in lipid accumulation [101].

For this polymorphism, subpopulation differences in allele frequency between the geographical groupings analyzed in the scope of the 1000 Genomes Project did not surpass 8% (except for American subpopulations) [102]. In addition, among 40 SNPs located in intron 1 that showed relationships with diseases based on GWAS, 35 SNPs exhibited less than 5% interpopulation variances in the recording frequency of one of the alleles. This can be explained by the fact that, in addition to significant population differentiation in SNP allele frequencies, large blocks of linkage disequilibrium were discovered in the region of the FTO intron 1 in European and Asian populations (blocks of linkage are smaller in African populations) [102,103]. Furthermore, association studies revealed that haplotypes could create variations with risk/protective effects on pathological conditions, including BMI and obesity.

In another study, researchers in Hungary found that for 11 SNPs, risk allele frequencies considerably changed across the two ethnic minorities: the Hungarian and the Roma [104]. Variants in the fat mass and obesity-associated (FTO) gene (rs1558902, rs1121980, rs9939609, and rs9941349) showed a robust yet ethnicity-independent link with obesity. The Roma community had greater connections with obesity than the Hungarian general population, which was explained by the ethnicity-associated behavioral and environmental factors. 

In a recent 2021 study by Boiko et al., four FTO SNPs were found to be significantly associated with body mass index in patients with schizophrenia, irrespective of the treatment regimen [105]. Furthermore, such association was not confirmed for antipsychotic drug-induced metabolic syndrome. It is not clearly understood if the presence of specific FTO SNPs will increase the risk of obesity in patients receiving antipsychotic medications.

### 4.2. Genes Associated with Metabolic Syndrome

#### 4.2.1. ADRA1A

Weight increases in people with schizophrenia have also been linked to a range of genetic variations. For instance, the alpha-1A adrenergic receptor (ADRA1A) gene has been linked to cardiovascular risk factors such as obesity and hypertension, and a positive connection has been observed between the presence of the Arg347 allele of ADRA1A and the total number of metabolic syndrome (MetS) components [106].

When the three genes for alpha1 adrenergic receptors (ADRA1A) in the human genome (ADRA1A, ADRA1B, and ADRA1D) were investigated in the population of the United States ADRA1A, ADRA1B, and ADRA1D, haplotype blocks of various lengths were detected in the Caucasian and African American populations [107]. 

Haplotypes exhibit substantial linkage disequilibrium over extended chromosomal areas [108]. Therefore, human genome haplotype-block organization can have crucial implications for successfully mapping genetic polymorphisms linked with complicated diseases. Once the haplotype blocks of a candidate gene have been identified, a collection of haplotype tag SNPs that can capture the haplotype variety of the blocks may be chosen. This offers an effective method for screening each haplotype block for association, particularly because they allow for the discovery of effects of any allele of moderate abundance and effect size, even if the causal allele is unknown [109]. In the U.S. Caucasian population, all SNP markers fell within haplotype blocks in ADRA1A and ADRA1D. In general, shorter haplotype blocks were found in African Americans, and 30–40% of the genomic regions of ADRA1B and ADRA1D did not exhibit block structure in this population [107].

The findings from the study confirm that the haplotype block architecture of three alpha-adrenergic receptor genes exhibit demographic disparities in haplotype block structure between Caucasians and African Americans in the United States.

#### 4.2.2. eNOS

Endothelial nitric oxide synthase (eNOS) produces NO in endothelial cells and platelets, and it is critical for maintaining vascular homeostasis, preventing platelet and leukocyte adhesion, and inhibiting vascular smooth muscle cell migration and proliferation. Moreover, clinical investigations have demonstrated that functional polymorphisms or haplotypes in the eNOS gene are linked to an increased risk of MetS [110,111].

In a study that aimed to see if eNOS gene polymorphisms or haplotypes are linked to MetS vulnerability in children and adolescents, the distribution of genetic variations of three clinically important eNOS polymorphisms (T786C in the promoter, VNTR in intron 4, and Glu298Asp in exon 7) in ethnically-defined DNA samples was examined to calculate the haplotype frequency and look for correlations between these variations [112]. 

In [113], Caucasians (34.5%) had a higher prevalence of the Asp298 variation than African Americans (15.5%) or Asians (8.6%), and a higher prevalence (42.0%) of the C-786 variation than African Americans (17.5%) or Asians (13.8%). African Americans (26.5%) had a higher prevalence of the 4a variation in intron 4 than Caucasians (16.0%) or Asians (12.9%). In each of the three groups, the most frequent projected haplotype exclusively included wild-type variations. This haplotype was more prevalent in Asians (77% vs. 46% in the other ethnicities). In African Americans, the second most frequent haplotype contained the variation 4a and wild-type variants; the Asp298 and 4a variants were negatively related in this group. 

Since the biological changes associated with the T786C polymorphism predispose children and adolescents to MetS, genetic tests should be performed to clinically address variants as a result of the aforementioned interethnic differences.

Further studies are required to decipher the potential effects of those gene variants in patients receiving psychotropic medications and whether they can contribute to an increased susceptibility and/or degree of severity of their metabolic adverse effects.

## 5. Impact of Variability in SNP Frequency among Ethnicities

### 5.1. CYP2D6 Isoenzyme

The polyallelic attribute of CYP2D6 has associated Caucasian populations with Poor Metabolizers (PM) (8%), of which Asian populations have a lower prevalence (1%). On the other hand, Intermediate Metabolizers (IM) are more prevalent among Asian populations (35–55%) than in Caucasians (<2%) (5, 8). Although the CYP2D6*4 allele contributing to the PM phenotype is more frequently found in Caucasians, population studies reveal that Asians harbor the highest frequency of the decreased function CYP2D6*10 allele (52%); white Europeans and Oceanians account for lower than 3–7% and it is seldom seen in African populations (Figure 2) [11].

Saruwatari et al. [115] analyzed the non-linear pharmacokinetic (PK) parameters of the Michaelis–Menten constant (Km) and maximum velocity (Vmax) in major depressive disorder Japanese patients who were prescribed paroxetine to investigate the effects of CYP2D6 polymorphisms, including CYP2D6*10, on plasma paroxetine concentrations. Results indicated a significant difference in the CYP2D6*10 carriers than non-carriers between the Kmax (24.2 ± 18.3 ng/mL and 122.5 ± 106.3 ng/mL, *p* = 0.008) and Vmax values (44.2 ± 16.1 mg/day and 68.3 ± 15.0 mg/day, *p* = 0.022), respectively. Owing to the inter-ethnic disparities in the CYP2D6*10 allele frequency, genotyping individuals could contribute to achieving optimal blood paroxetine concentrations.

### 5.2. Ethnic Variation of CYP2C19

Galindo et al. [82] compiled 138 research studies to classify the prevalence of CYP2C19 alleles based on ethnic groups and geographical regions. Compared to the rest of the population, the CYP2C19*2 allele was more widespread in Native Oceanians (61.3%), and thereafter, in East and South Asians (30.3%). The CYP2C19*3 allele was likewise more frequent in Native Oceanians (14.42%) and East Asians (6.89%); in contrast, it was infrequent in the rest of the ethnic groups. Moreover, the star 17 allele was more prevalent in the Mediterranean and South Europeans (42%), and in the Middle Eastern region (24.87%). Frequencies are listed below from PharmGKB [82] in Figure 3.

Although the non-functional CYP2C219*2 and *3 are the most commonly genotyped alleles, extensive and ultra-rapid metabolizer phenotypes of CYP2C19*17 exhibit the highest inter-ethnic diversity [29]. Clinically, such phenotypes have a notable impact on CYP2C19 substrates, such as amitriptyline, a tricyclic antidepressant that works as both a serotonin and norepinephrine reuptake inhibitor [116]. Kirchheiner et al. [32] reported treatment with antidepressants such as amitriptyline, among others, would benefit from the CYP2C19-based dose adjustment by reducing the drug dosage prescribed by 110% for carriers of homozygous Extensive Metabolizer (EM) allele, <100% for heterozygous and 60% for PM. Owing to the vast inter-ethnic differences between CY2C19*17 (Figure 3) [82], individualized dosing would significantly impact the therapeutic response. 

### 5.3. Ethnic Variation of CYP2C9

Among the two most common variants are the decreased function CYP2C9*2 and the no-function CYP2C9*3. The CYP2C9*2 allele has frequencies ranging from 11–13% in Middle Eastern, European, and South/Central Asian populations; however, it has an estimate of 2% in African ancestry and <1% in East Asian populations. CYP2C9*3 ranges from frequencies of around 7–11% in European populations, Middle Eastern, and South/Central Asian ancestry, but is lower in East Asian (3%) and is even lower in African populations (Figure 4) [117].

In a study by Zubiaur et al. [84], 80 participants were profiled for pharmacokinetic parameters through blood sample collection pre-dose and up to 72 h after olanzapine intake, and were subsequently genotyped. Analysis revealed PM were linked to statistically significant higher half-life (t ½) and volume of distribution compared to Normal Metabolizers (NM) or IM. This showed the accumulation of olanzapine to a wider degree than the other phenotypes. In addition, polymorphism was related to adverse drug reactions and the PK variability was congruent with the polymorphism of transporters. 

Since PMs of homozygous for CYP2C9*3 or heterozygous CYP2C9*3/*4, among others, could lead to such consequences, correlating inter-ethnic differences with allelic function emphasizes the potential need for dose adjustment and possible prevention of adverse effect if the inter-ethnic differences are considered during prescription by physicians. 

### 5.4. Ethnic Variation of HLA (Human Leukocyte Antigen)

When compared to Japanese (0.002), Koreans (0.004), and Europeans (0.01–0.02), the carbamazepine-induced Stevens-Johnson syndrome/toxic epidermal necrolysis (SJS/TEN) was comparatively higher among Han Chinese (0.057–0.145), Malaysians (0.12–0.157), and Thai (0.085–0.275) [118,119,120,121,122,123,124]. More recent genome-wide association studies (GWAS) have revealed that the HLA-A*31:01 allele has a relatively stronger association with carbamazepine-induced hypersensitivity in the populations with lower frequency of HLA-B*15:02, namely Northern Europeans, Japanese, and Koreans [118,125,126,127]. Furthermore, the HLA-B*15:11 allele has also been linked to carbamazepine-induced SJS/TEN in Japanese and Korean populations [118,124]. Pharmacogenetic studies in East Asians from Taiwan, Thailand, and Japan found that phenytoin-related ADRs are linked with CYP2C9*3 and HLA-B*13:01, HLA-B*15:02, and HLA-B*51:01 [128,129]. Moreover, similar phenytoin-related ADRs have been reported to be elevated in Thai and Malay patients with the HLA-B*13:01, HLAB*56:02/04, and CYP2C19*3 variants, or when omeprazole is co-administered in patients of Chinese descent [130,131]. The HLA-A*02:01:01, HLA-B*35:01:01, and HLA-C*04:01:01 haplotypes have also been shown as biological markers for lamotrigine-induced ADRs in Mexicans [132]. 

## 6. Non-Genetic Factors Influencing Cytochrome P450s Activity

Extrinsic factors, such as smoking, pregnancy, age, and the use of concomitant medications, interact with CYP450 and potentially affect their catalytic activity. Cigarette smoking can potently induce the expression of the inducible enzyme CYP1A2; however, the induction effect depends mainly on the CYP1A2 genotype [133,134]. For instance, phenoconversion into a faster metabolizing phenotype was investigated in CYP1A2*1F smokers; consequently, higher doses of olanzapine were needed for them to have an effective olanzapine plasma concentration. Additionally, other drugs, such as clozapine, require analyzing variant polymorphisms of CYP2D6, CYP2C19, and CYP1A2, along with several nongenetic factors, specifically smoking and concomitant medications [135]. The inhibition by other drugs is also a problem that impairs the genetic data, thus changing the apparent phenotype. For example, receiving carbamazepine and valpromide in combination leads to an increase in the patient’s carbamazepine plasma level, since EPHX1 metabolizes carbamazepine and, at the same time, is inhibited by valpromide [136].

Aging is another critical factor that alters CYP450 genes expression, and consequently, their proteins’ metabolic functionality. Notably, age can decrease biotransformation activity of CYP450 enzymes, thus leading to a low metabolizer phenotype. Similarly, genetic variations in addition to the demographic parameters have been investigated to contribute to the modified CYP2B6 expression and activity [134,137]. Although hepatic CYP2B6 represents only a small proportion (1–4%) of the human CYP450, it is responsible for catalyzing the metabolism of many important drugs, including the atypical antidepressant bupropion [138]. 

Regarding pregnancy as a nongenetic factor, conflicting results for the impact of pregnancy on CYP2D6 enzymatic activity from two different studies on pregnant women receiving paroxetine and dextromethorphan have been obtained. Particularly, for CYP2D6 PMs and CYP2D6 IMs, contradictory results were reported, most probably due to the metabolism of drugs by alternative CYP450 enzymes “other than CYP2D6” that exhibit low enzymatic activity specifically during pregnancy [134]. 

Overall, single nucleotide polymorphisms (SNPs) play a major role in inter-individual variability in therapeutic drug response. In contrast, in a particular genotype group, variability is still observed, suggesting the contribution of nongenetic factors that influence CYP450 activity [139]. 

## 7. Pharmacogenomics and the MENA Region

In a comprehensive analysis, Jithesh et al. [140] investigated the population of Qatar. A total of 6,045 whole genomes from Qataris revealed 1320 variants in 703 genes that ultimately affect 299 drugs, and were significantly different from those from the other populations (76, 156 whole genomes) archived in the gnomAD v3 dataset.

Furthermore, 615 of the variants were more frequently found in Qataris. The rs1137101 SNP in the LEPR gene was found to be lower in the Qatari population. The LEPR gene encodes for the leptin receptor and mutations were associated with obesity and pituitary malfunction [141]. On the other hand, the rs2289669 in SLC47A1 and rs11212617 in ATM were both higher. The former encodes the multidrug and toxin extrusion protein 1 (MATE1) and has been linked with cellular uptake and sensitivity to the anticancer drug imatinib [142], while mutations in the latter cause the neurodegenerative disease of ataxia-telangiectasia [143]. Moreover, an assessment of 15 pharmacogenes that impact 46 medications revealed individuals were found to possess 3.6 actionable genotypes/diplotypes on average, with at least one clinically actionable genotype/diplotype seen in 99.5% of the study participants. Moreover, on average, Qataris were found to carry pharmacogenetic variants that predict actionable phenotypes, which ultimately influence 12.9 (28.8%) of the 46 medications. 

Additionally, a genome-wide association (GWAS) study (n = 182) reported the genetic variations related to the weekly warfarin dosing requirements in Middle Eastern and North African (MENA) populations. Results revealed variants rs9934438 within Vitamin K Epoxide Reductase Complex Subunit 1 (VKORC1) and rs4086116 in CYP2C9 accounted for 39% and 27% of the variability seen in the Qatari (n = 132) and Egyptians (n = 50), respectively [144]. In the endoplasmic reticulum membrane, VKORC1 produces the catalytic component of the vitamin K epoxide reductase complex, which converts inactive vitamin K 2,3-epoxide to active vitamin K. Therefore, allelic variation could cause increased sensitivity or resistance to warfarin, a vitamin K epoxide reductase inhibitor [145]. 

In another study conducted in association with the SEAPharm consortium project [146], Al-Mahayri et al. [147] analyzed the landscape of variation among the indigenous citizens of the United Arab Emirates through targeted resequencing. The DNA from 100 self-identified Emirati participants was extracted from whole blood samples and was resequenced by utilizing the targeted sequencing panel (PKSeq). The aim of the study signifies that despite the fact rare variants (allele frequency <1%) are seldom observed in clinical trials, many of them have been determined to be actionable. Such variants “cluster geographically” or become exclusive to a population, which necessitates the creation of libraries archiving rare variants around the world [148].

In population genetics, a genetic variant is characterized as common if its minor allele frequency (MAF) is higher than 1%, and uncommon if its MAF is less than 1%. Out of the 1243 variants identified, a majority (63%) of these variants were found in a MAF less than or equal to 1% (MAF ≤1%). Interestingly, when compared to other populations from multiple databases, around 30% of the identified variants in the Emirati participants were unique [149].

Moreover, among the pharmacogenes investigated in this study, the CYP family had the largest number of variations. Within this family, the CYP2D6 gene came second to CYP4F12, which bore the greatest number of variants. From the CYP2D6 variants, eleven were identified as key markers in clinically actionable haplotypes. Thirty participants possessed CYP2C9*2 or *3 alleles (low and no function, respectively), while three were homozygous for one of the aforementioned alleles.

The highest detected diplotype was the CYP4F2*1/*3 (54%), while CYP2B6*1/*18, CYP2C9*2/*3 (1%) were the lowest among the CYP family. Moreover, although the detected SNP for ABCC4 had the highest allele frequency (rs1751034, 82.6%), among the CYP family identified in the current group, in a descending order of allele frequency, CYP4F2 had the highest allele frequency (rs2108622, 45.92%), followed by CYP2B6 (rs3745274, 31%), CYP2C19 (rs4244285, 15.1%), CYP2C8 (rs10509681, 13.64%), CYP2C9 (rs1057910, 7.07%), CYP2C8 (rs7900194, 2%), and CYP2B6 (rs28399499, 1.01%) [147].

Such studies highlight the significance of pharmacogenetics research, particularly in the Middle East, where the lack of such data and awareness hinders the implementation of pharmacogenomics into practice [150].

## 8. Pharmacogenomics in Real World of Psychiatry Practice

The CPIC guidelines are good tools by which to recommend the appropriate selection and dosing of a variety of medications, with those pertaining to psychiatry the most prominent. To adopt an approach of pharmacogenomics-guided clinical practice, genotyping results and consequent recommendations should be available at the point of care. Different levels of metabolism are assigned to different gene variants (e.g., intermediate metabolizer, poor metabolizer, etc.). However, the development of activity scores can provide a better classification of different grades of metabolism with complexity; whereas medications are metabolized by more than one enzyme. Notably, the activity score may depend on the medication itself (substrate dependency) [151]. 

There have been several examples in which enzymatic activity markedly varies according to genotype, with potential consequences on drug efficacy and safety. Previous studies investigated more than 2000 patients receiving escitalopram by CYP2C19 genotyping patients. The *17 allele increased the enzymatic capacity of CYP2C19 by only approximately 20% compared to the wild type [152]. In another example, in more than 1000 patients receiving venlafaxine and genotyped for CYP2D6, patients with *9 and *10 alleles had a 70% reduction of enzymatic capacity; whereas, the presence of *41 allele reduced the enzymatic capacity by around 85% [153]. To add to this complexity, as previously discussed, enzymatic activity may vary in different ethnicities, and therefore, in vivo validation studies of such predictive values are required. Moreover, many patients may have comorbidities and receive several medications, with potential drug interactions. Accordingly, mathematical modeling may help integrate numerous factors affecting drug response, thus optimizing the selection and dosing of medications. 

Integrating the results of genotyping into electronic medical records is another challenge to the routine use of pharmacogenomics in clinical practice. The recommendations should also be integrated in order to provide adequate guidance to the physician at the site of care. The cost of genetic testing should be interpreted within the context of total (direct and indirect) costs of mental health disorders. Notably, indirect costs are high for psychiatric patients [151], whereas the cost of genetic testing has markedly dropped by using new technologies and high throughput devices. 

One of the important causes of non-compliance is the development of adverse drug reactions. As previously described, adding HLA genotyping and predictors of obesity and metabolic syndrome may help precise selection of medications, and avoid prescribing those with a high probability of causing adverse drug reactions in patients possessing certain gene variants. 

## 9. Challenges

Psycho-pharmaco-genetic studies seem to concur with the effect genetic variations have on the treatment outcomes for patients on psychotropic medications and can be utilized to optimize patient therapy. However, thus far, much of the pharmacogenetic studies have been contradictory. For instance, while CPIC advises CYP2D6 poor metabolizers to avoid amitriptyline [31], the FDA simply discusses its use in the same phenotypic population as a warning [49]. This has contributed to hindering the implementation of pharmacogenomics into practice, with the lack of prescriber training and confidence in interpreting the results being additional contributing factors [154,155,156,157]. Moreover, the contradictory results of psychiatric genotyping studies, due to small sample size, lack of psychopharmacological expertise, and demographic, clinical and environmental differences in patient cohorts, have limited the utilization of psychopharmacologic assays to specific psychotic disorders, such as refractory schizophrenia [59]. Furthermore, the complexity of psychiatric diseases and the inter-ethnic differences in drug response is a challenge for pharmacogenomics testing in psychiatric clinics [158].

Shugg, T. et al. [159] investigated inconsistencies with pharmacogenomic recommendations from the major U.S. reporting sources (CPG, CPIC, and FDA), which significantly differed in the following categories: recommendation, addressing routine screening, specific biomarkers, variants, and patient groups. The study concluded that almost half of the recommendations were inconsistent. Proposed causes included: (1) inconsistencies between the sources in the level of evidence required to deem the drug-gene pair as “clinically actionable”; (2) different mission statements between the sources (i.e., where the CPIC recommends, and the FDA provides information rather than recommendations).

Potential solutions, as reported by Shugg et al., include constructing strong evidence that favors pharmacogenomics generated through randomized controlled trials, the collaboration of expert panels with official organizations and consortia, or collective agreement of organizations to follow a single source for pharmacogenomic recommendations [159]. 

Moreover, as previously described, genetic background significantly differs among different ethnicities. The regional disparities in allele frequencies could complicate interpretations of pharmacogenomic results among the various groups of subjects involved in the studies. Further complexity arises from the ancestral origin of the said groups. For example, North Indians, South Indians, and East Indians are all genetically distinct, but are grouped into a “homogenous” Asian population [160]. Moreover, self-reported ethnicities can cause complications in genetic dosing algorithms. Additionally, doubts have recently been raised regarding the necessity for race-based algorithms [161], as it is unclear what ethnic and racial groups should be used as a standard in such studies, with paucity of studies in many ethnicities, including those of the Middle East [162]. This calls for the need to establish well-defined categories than the broader black, white, and Asian, to be able to use ethnicity as a proxy when, for example, no information is available for a patient’s genotype, as well as to improve precision in pharmacogenetics research.

## 10. Conclusions

The “trial-and error” method of selecting psychotropic agents for psychiatric patients implores the need to find ways that improve the current prescribing patterns for patients. In this review, we have demonstrated the recent significant progress in our understanding of individualized therapy. Data showed that, based on the inter-differences in drug response, pharmacogenomic studies in clinical practice can aid in identifying reasons behind an individual’s lack of response and/or the occurrence of an adverse drug reaction. However, disparities in technological advancements and research capacities between regions around the world have created a paucity of data on the prevalence of actionable pharmacogenomic variations in several countries, including the Middle East. Moreover, the complex genetics and phenotypes of psychiatric diseases, as well as the apparent inter-ethnic differences, minimize the utilization of pharmacogenomics testing in psychiatric clinics. 

In addressing the above-mentioned challenges, future work should focus on transforming the world towards practicing precision medicine based on individualized genetic profiles. For this purpose, it is essential that healthcare professionals have sufficient knowledge of genetic principles. Practically, increased awareness toward the significance of allelic variants in clinical practice and establishment of electronic genetic records can prospectively identify patients that would benefit from pharmacogenetic testing, and thus, appropriate psychiatric treatments.

## Figures and Tables

**Figure 2 ijms-23-13485-f002:**
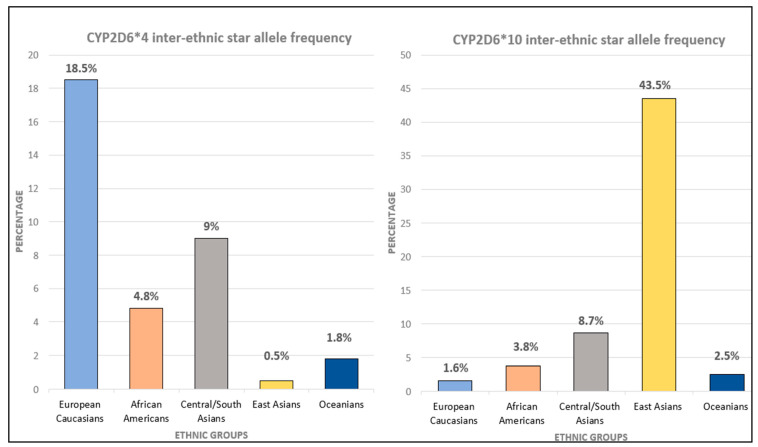
CYP2D6 inter-ethnic star allele frequency (https://www.pharmgkb.org/page/cyp2d6RefMaterials, PharmGKB) accessed on 8 February 2022, [114].

**Figure 3 ijms-23-13485-f003:**
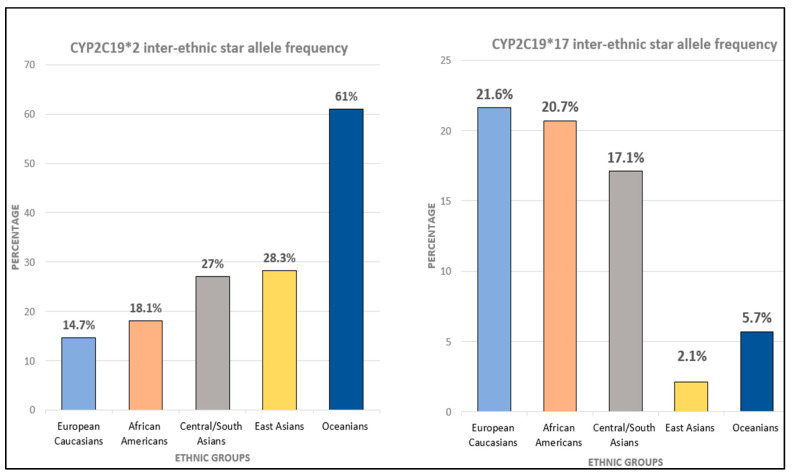
CYP2C19 inter-ethnic star allele frequency (https://www.pharmgkb.org/page/cyp2c19RefMaterials, PharmGKB), accessed on 8 February 2022 [82].

**Figure 4 ijms-23-13485-f004:**
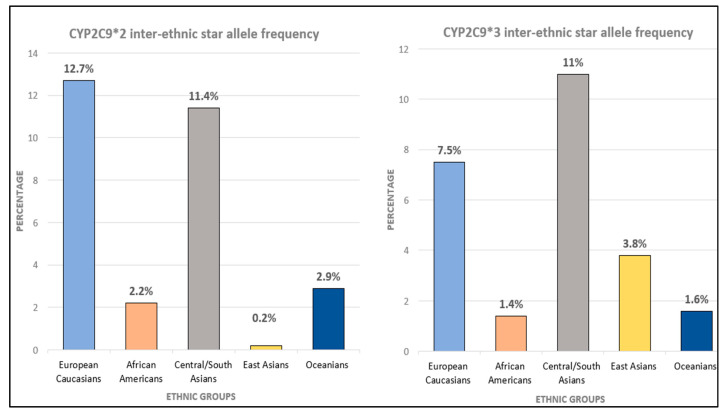
CYP2C9 inter-ethnic star allele frequency (https://www.pharmgkb.org/page/cyp2c9RefMaterials, PharmGKB), accessed on 8 February 2022 [80].

**Table 3 ijms-23-13485-t003:** Dosing recommendations for psychotropic drugs based on CPIC guidelines for selected CYP and HLA-B phenotypes.

Enzyme	Star Allele (Genotype)	Diplotype Examples	Metabolizer (Phenotype)	CPIC ^®^ Guidelines	Reference
CYP2D6	*4, *10	*4/*41 *4/*10 *10/*10 *10/*41	Intermediate	Dosing recommendations for TCAs: Reduce recommended starting dose by 25%. Guide dose adjustments via therapeutic drug monitoring (TDM). Dosing recommendation for paroxetine: Commence therapy with recommended starting dose. Dosing recommendation for fluvoxamine: Commence therapy with recommended starting dose.	[31,78,79]
		*4/*4	Poor	Dosing recommendations for TCAs: Avoid tricyclic use because of the propensity for side effects. Consider alternative drug not metabolized by CYP2D6. If a TCA used is called for, a 50% reduction of recommended starting dose should be considered. Guide dose adjustments via therapeutic drug monitoring (TDM). Dosing recommendation for paroxetine: Choose an alternative drug not primarily metabolized by CYP2D6. If paroxetine use is warranted, reduce recommended starting dose by 50% and titrate to response. Dosing recommendation for fluvoxamine: Reduce recommended starting dose by 25–50% and titrate to response. Consider use of an alternative drug not metabolized by CYP2D6.	
CYP2C9	*2, *3	*1/*2 *1/*3 *2/*17 *3/*17	Intermediate	Dosing recommendation for phenytoin/fosphenytoin based on HLA-B*15:02: (1) First dose: usual loading dose. (2) Subsequent doses: 25% less than normal maintenance dose. (3) Adjust subsequent dose based on TDM.	[80,81]
		*2/*2 *3/*3 *2/*3	Poor	Dosing recommendation for phenytoin/fosphenytoin based on HLA-B*15:02: (1) First dose: usual loading dose. (2) Subsequent doses: 50% less than normal maintenance dose. (3) Adjust subsequent dose based on TDM.	[80,81]
CYP2C19	*2, *17	*17/*17	Ultra-rapid	Dosing recommendations for the tertiary amines: Avoid tertiary amine use because of the likelihood for sub-optimal response. Consider alternative drug not metabolized by CYP2C19. If a tertiary amine use is warranted, guide dose adjustments via therapeutic drug monitoring (TDM). Dosing recommendations for citalopram and escitalopram Choose an alternative drug not primarily metabolized by CYP2C19. Dosing recommendations for sertraline Commence therapy with recommended starting dose. If patient fails to respond to recommended maintenance dosing, choose an alternative drug not primarily metabolized by CYP2C19.	[31,82]
		*1/*17	Rapid		
		*1/*2 *2/*17 *3/*17	Intermediate	Dosing recommendations for the tertiary amines: Commence therapy with recommended starting dose. Dosing recommendations for citalopram and escitalopram 3. Commence therapy with recommended starting dose. Dosing recommendations for sertraline Commence therapy with recommended starting dose.	

**Table 4 ijms-23-13485-t004:** Alternative drugs to consider when following CPIC guidelines for selected psychotropics and CYP isoenzymes.

Enzyme	Selected Psychotropic(s) Affected by Enzyme	Alternative Psychotropics	Reference
CYP2D6	Tricyclic antidepressants	Secondary amines, Desipramine Nortriptyline.	[79]
CYP2C9	Olanzapine	Asenapine, Quetiapine.	[83,84,85,86]
	Phenytoin Fosphenytoin	Eslicarbazepine, Lamotrigine, Phenobarbital.	
CYP2C19	Sertraline Escitalopram Citalopram	Fluoxetine, Fluvoxamine, Paroxetine.	[31,83]

## Data Availability

Not applicable.

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
