# Peer review of "Pharmacogenomics in Psychiatry Practice: The Value and the Challenges"

_ijms, 2022, doi:10.3390/ijms232113485_

Round 1

Reviewer 1 Report

Authors analyse the role of P450 enzymes in psychiatric practice, largely including P450 CYP polymorphisms and HLA polymorphisms data. This argument, connected with drug pharmacogenomics, is controversial in the literature, nevertheless it is interesting and the present review is welcomed.

Major comments:

1.                  Due to the fact that abundant part of this review is dedicated to HLA variants, it need to be reflected in the tittle. Thus, the title change is recommended.

2.                  Chapter 4 is entirely dedicated to obesity and metabolic syndrome, with apparently small connection with psychiatry. Obesity and metabolic syndrome are not mentioned yet in the Abstract: please, update the Abstract. Please, connect them better with psychiatry in the text.

Minor comments:

1.                  In the author affiliation the country is not indicated. The same in the Funding section (line 636): the country is missing.

2.                  Line 89. Authors list some CYP enzymes as metabolisers, and for anticonvulsants CYP2C9, HLA.A and HLA-B are listed. Why HLA-A and HLA-B are listed as enzymes?

3.                  Fig 1. In Figure body, “OXV”, “Gol” and other abbreviations are present, but they are not defined/explained in the figure Legend.

4.                  Line 78, in the text, the data from Fig 1 are referred to [16], but in the figure legend, line 112, the reference is [24]. Fix it, please.

5.                  Table 2. The horizontal row separator lines are welcomed. Now is difficult to distinguish which drug belongs to which CYP.

6.                  Line 212, the term “biochemical metastasis” is not clear. Better define it or change it.

7.                  Table 3. The horizontal row separator lines are strongly welcomed.

8.                  Table 3. CYP and HLA phenotypes are described in the Table, but HLA are not mentioned in the Table title, only CYP. Please add HLA.

9.                  Line 282. It is not clear, why in the text the percentage of CYP2D6*10 allele in Asians are indicated as 64% with reference to the Figure 2, but in the Figure 2 the column is signed as 43,5%?

10.              Fig 2 and Fig 3 legends: “pharmgkb” is more correct as PharmGKB. The web link is welcome.

11.              Line 287. Please, check the correctness of the phrase “Saruwatari et al. (88) analyzed the non-linear pharmacokinetic (PK) parameters of Michaelis–Menten constant (Km) and maximum velocity (Vmax) in major depressive disorder Japanese patients…”  The pharmacokinetic of enzymatic constants is not correct.

12.              Fig 3: in the text the reference to the Figure 3 is completely missing.

13.              Line 339. Spell the acronym SJS/TEN when the first time mentioned.

14.              References: in some references number of volume or even journal name is missing. Eg. Ref 3, 4, 14, 20, 33, 149.

15.              In some references, the web link is welcome: ref 21, 55.

Author Response

The authors would like to thank the reviewer for the valuable comments that helped us to substantially improve the manuscript. Accordingly, we have thoroughly revised the manuscript, edited and  corrected grammar or spelling errors. We improved the writing flow and the language, as well as the link to the introduction and different sections. We updated and completed reference list. Kindly find below our response to each comment. 

Major comments:

  1. Due to the fact that abundant part of this review is dedicated to HLA variants, it need to be reflected in the tittle. Thus, the title change is recommended.

Authors’ response: We thank the reviewer for the valuable comment. We have changed the title to be “Pharmacogenomics in Psychiatry Practice: The Value and the Challenges”, to cover variants beyond those of CYP450.

  1. Chapter 4 is entirely dedicated to obesity and metabolic syndrome, with apparently small connection with psychiatry. Obesity and metabolic syndrome are not mentioned yet in the Abstract: please, update the Abstract. Please, connect them better with psychiatry in the text.

Authors’ response: We thank the reviewer for the valuable comment. The following were added to justify the addition of this section:

Abstract was updated and we referred to obesity and metabolic syndrome-related gene variants Lines #17-20. 

Introduction: Lines# 64- #66: “Finally, we will discuss the genetic susceptibility to obesity and metabolic syndrome, as both are recognized adverse effects of the second generation anti-psychotic medications”.

The susceptibility to obesity and metabolic syndrome, as an adverse effect of some psychiatric medications, may be predicted by the presence of specific gene variants.

Section 4: Lines #316-318 describes the justification of adding obesity- and metabolic syndrome-related genes to the pharmacogenomics panel to be recommended in psychiatry patients

Minor comments:

  1. In the author affiliation the country is not indicated. The same in the Funding section (line 636): the country is missing.              Authors'response: Thanks for the comment. We added the country to both.

  2. Line 89. Authors list some CYP enzymes as metabolisers, and for anticonvulsants CYP2C9, HLA.A and HLA-B are listed. Why HLA-A and HLA-B are listed as enzymes?                                           Authors'response: Corrected by removing HLA.A and HLA-B in this line. 
  3. Fig 1. In Figure body, “OXV”, “Gol” and other abbreviations are present, but they are not defined/explained in the figure Legend.      Authors'response:Corrected. Explained in the figure legend (Illustration provides an insight into correlation between binding sites named by their unique ligands (OXV: (4-hydroxy-3,5-dimethylphenyl)(2-methyl-1-benzofuran-3-yl)methanone, GOL: glycerol ,HEM: protoporphyrin ix containing Fe, HEC: heme c and SWF: s-warfarin) and genome variations (defined as dbSNP number) deposited in PDB (retrieved from Protein Data Bank)
  4. Line 78, in the text, the data from Fig 1 are referred to [16], but in the figure legend, line 112, the reference is [24]. Fix it, please.                              Authors'response:Thanks for your valuable comments. There were missing references in the figure legend. We added the correct references.
  5. Table 2. The horizontal row separator lines are welcomed. Now is difficult to distinguish which drug belongs to which CYP.

       Authors'response: Corrected. horizontal row separator lines have been added. 

6. Line 212, the term “biochemical metastasis” is not clear. Better define it or change it.

Authors'response: Corrected. By replacing it with cellular homeostasis which means CYP450 play a pivotal role in cellular metabolism and homeostasis. 

7.Table 3. The horizontal row separator lines are strongly welcomed.

Authors'response: 

We have edited the tables throughout by adding borders on all four sides of the tables.

8. Table 3. CYP and HLA phenotypes are described in the Table, but HLA are not mentioned in the Table title, only CYP. Please add HLA.

Authors'response: 

Thank you for pointing this out. We have added “HLA-B phenotypes” to the table title.

9. Line 282. It is not clear, why in the text the percentage of CYP2D6*10 allele in Asians are indicated as 64% with reference to the Figure 2, but in the Figure 2 the column is signed as 43,5%?

Authors'response: 

We thank you for pointing out the inconsistency. The percentage in the text had been obtained from another reference, that cited the original article by Qin et al., (Systematic polymorphism analysis of the CYP2D6 gene in four different geographical Han populations in mainland China”, 2008) and the figure was constructed from values in the PharmGKB database. The percentage was added to provide the maximal value that has been recorded in literature regarding the allele frequency in Asians. However, to prevent any confusion and be consistent with the text and figure, we have omitted 64% and replaced it with “52%” rounded to the nearest whole number, to refer to the percentage harbouring the CYP2D6*10 allele in Asians altogether (including East Asians, central and south Asians), from PharmGKB.

10. Fig 2 and Fig 3 legends: “pharmgkb” is more correct as PharmGKB. The web link is welcome.

Authors'response: Thank you for pointing this out. The document has been updated to replace the terms pharmgkb to the more accurate “PharmGKB”. Additionally, the weblinks have been included in lines 286, 306 and 338 to provide direct access to the respective allele frequency tables.

11. Line 287. Please, check the correctness of the phrase “Saruwatari et al. (88) analyzed the non-linear pharmacokinetic (PK) parameters of Michaelis–Menten constant (Km) and maximum velocity (Vmax) in major depressive disorder Japanese patients…”  The pharmacokinetic of enzymatic constants is not correct.

Authors'response:

Thank you for pointing out the confusion in the text. Instead of including only the P values to highlight the significant differences, we have replaced (P=0.008) with (24.2±18.3 ng/mL and 122.5±106.3 ng/mL, P=0.008) and (P=0.022) with (44.2±16.1 mg/day and 68.3±15.0 mg/day, P=0.022). The values represent the following: (Kmax of carriers and non-carriers, P value) and (Vmax of carriers and non-carriers, P value), respectively.

12. Fig 3: in the text the reference to the Figure 3 is completely missing.

Authors'response: Reference to Figure 3 was added in the text: Lines #453 and # 465.

13. Line 339. Spell the acronym SJS/TEN when the first time mentioned.

Authors'response: corrected

14. References: in some references number of volume or even journal name is missing. Eg. Ref 3, 4, 14, 20, 33, 149.

Authors'response:

Data for Ref #3 and #4 were completed.

Ref #14 was replaced

Ref #20 was completed

Ref #33 was replaced by #14

Ref #149 was completed

15. In some references, the web link is welcome: ref 21, 55.

Authors'response:

-We changed reference 21:

Gitlin M. Lithium side effects and toxicity: prevalence and management strategies. Int J Bipolar Disord. 2016 Dec;4(1):27. doi: 10.1186/s40345-016-0068-y. Epub 2016 Dec 17. PMID: 27900734; PMCID: PMC5164879.

-We added the web link to reference #55:

Bipolar disorder: assessment and management. London: National Institute for Health and Care Excellence (NICE); 2018 Apr. (NICE Clinical Guidelines, No. 185.) Available from: https://www.ncbi.nlm.nih.gov/books/NBK547001/

-We checked all other references

Best Regards

(Corresponding Author)

Reviewer 2 Report

In their paper, authors review the role of CYPp450 and it's relative importance in psychiatry. They report effects of CYPs variants and non genetic factors on their metabolic activity and their role on some neurotransmitters' metabolism.

While the information provided are sufficiently described, I think there are some major flaws in some of the paper' sections. In details, the paper is titled <<Cytochrome P450 in Psychiatry Practice: The value and the Challenges>>, clearly indicating cyp450 as the central focus of the work. Why authors decided to include sections (apparently) not related to cyp450?

In details:

Lane 89 + Lane 337 and followings. HLA-A and HLA-B (major histocompatibility complex, class I, A, and B)  are proteins involved in immune system functions. although it was proved that they are risk factors for several side effects after drug treatment, they are not related to CYPs. Why including HLA if the focus is on Cytochrome P450 in Psychiatry Practice?

Lane 353 and following (Section 4): Authors describe Gene Variants Predicting Predisposition to Obesity and Metabolic Syndrome but there are no (apparent) links with CYP functions. It appears out of context.

Lane 455 and following (Section 5): while interesting, this section also appears out of context.

Lane 518 - 529: no links with cyp activity

I'd suggest to rimodulate these sections to clearly explain why the data reported is important in relation to CYP450 function. Alternatively, the sections can be removed.

Additionally, I'd like the authors to further discuss the potentiality of using CYPs genetic testing for individualized treatment. In authors' opinion what is the value of including CYPs genetics in clinical setting (in the choice of drugs), how they would use the data (as it is?, included in some predictive models?, which are the current limits of such approaches?).

Other than that, there are some really minor points mainly related to english language and style. I suggest a spell check. 

Line 71- 72. Please adjust the use of brackets

Line 75 mutations are contributed to > mutations are caused by

Lane 91 is directly eliminated > are directly eliminated

Table 1 CYP2C19 Ormal > Normal

Lane 125 repetition of for example.

Lane 158: Ref 42 reported link (http://bioinformatics.charite.de/transformer) is not accessible anymore. i think it was substituted by SuperCYPsPred by the same group PMID: 32182358

Lane 177 HRT2C > HTR2C

Lane 194 > encoded is correct? 

Lane 195 > however should be removed

Lane 199 concertation > concentration

Lane 207 "role" should be removed

Lane 207-208 plural should be used as multiple genes likely influence lithium response.

Lane 208 The reference 41 may be wrong here. there was no information on lithium

Lane 212 Metastasis is correct?

Lane 221 fatal > fetal

Lane 221-225 convoluted phrase. please simplify. 

Lane 230-232 while interesting as potential biomarker of cyp2d6 metabolic state, i feel the part regarding platelets could be moved on paragraph 2.2 from 2.(as paragraph 2. is mainly focused on  brain).

Author Response

The authors would like to thank the reviewer for the valuable comments that helped us to substantially improve the manuscript. Accordingly, we thoroughly revised the manuscript and improved the link between different sections. Kindly find below our response to each comment.

Major Comments:

Lane 89 + Lane 337 and followings. HLA-A and HLA-B (major histocompatibility complex, class I, A, and B)  are proteins involved in immune system functions. although it was proved that they are risk factors for several side effects after drug treatment, they are not related to CYPs. Why including HLA if the focus is on Cytochrome P450 in Psychiatry Practice?

Authors’ response: We thank the reviewer for the valuable comment. We apologize for this mistake, in fact, we kept an old title of the review draft by mistake. Now, with the updated title of the review, we are discussing all useful pharmacogenomics testing in psychiatry practice; including HLA.

Lane 353 and following (Section 4): Authors describe Gene Variants Predicting Predisposition to Obesity and Metabolic Syndrome but there are no (apparent) links with CYP functions. It appears out of context.

Authors’ response: We thank the reviewer for the valuable comment. We apologize for this mistake, in fact, we kept an old title of the review draft by mistake. Now, with the updated title of the review, we are discussing potential pharmacogenes that can guide psychiatry practice, including those which may predict higher incidence of obesity/metabolic syndrome with use of antipsychotic medications.

Lane 455 and following (Section 5): while interesting, this section also appears out of context.

Lane 518 - 529: no links with cyp activity

I'd suggest to rimodulate these sections to clearly explain why the data reported is important in relation to CYP450 function. Alternatively, the sections can be removed.

Authors response: We thank the reviewer for the valuable comment and suggestion.

Section 5 shifted to be now section 3: Lines #290-292 describe the justification of adding HLA to the pharmacogenomics panel to be recommended in psychiatry patients.

Section 4: Lines #316-318 describes the justification of adding obesity- and metabolic syndrome-related genes to the pharmacogenomics panel to be recommended in psychiatry patients.

Additionally, I'd like the authors to further discuss the potentiality of using CYPs genetic testing for individualized treatment. In authors' opinion what is the value of including CYPs genetics in clinical setting (in the choice of drugs), how they would use the data (as it is?, included in some predictive models?, which are the current limits of such approaches?).

Authors’ Response:

Thanks for the valuable comment. A new section was added to describe the pharmacogenomics in real world practice (section #8: Lines #725-759).

Minor Comments:

Line 71- 72. Please adjust the use of brackets

Authors' Response: Corrected. (Allelic variants of CYP enzymes, are commonly named according to the (*) system and translated to different phenotypes)

Line 75 mutations are contributed to > mutations are caused by

Authors’ Response: Corrected

Lane 91 is directly eliminated > are directly eliminated

Authors’ Response: Corrected

Table 1 CYP2C19 Ormal > Normal

Authors’ Response: Corrected

Lane 125 repetition of for example.

Authors’ Response: Corrected.

Lane 158: Ref 42 reported link (http://bioinformatics.charite.de/transformer) is not accessible anymore. i think it was substituted by SuperCYPsPred by the same group PMID: 32182358

Authors’ Response:  Corrected. Adding the following reference:

Carrascal-Laso L, Isidoro-García M, Ramos-Gallego I, Franco-Martín MA. Review: Influence of the CYP450 Genetic Variation on the Treatment of Psychotic Disorders. J Clin Med. 2021 Sep 21;10(18):4275. doi: 10.3390/jcm10184275. PMID: 34575384; PMCID: PMC8464829.

Line 177 HRT2C > HTR2C

Authors’ Response:  Corrected.

Lane 194 > encoded is correct? 

Authors’ Response: Corrected.

Replaced with catalyzed

Lane 195 > however should be removed

Authors’ Response: Corrected.

Lane 199 concertation > concentration

Authors’ Response: Corrected.

Line 207 "role" should be removed, Replaced gene with genes 

Authors’ Response: Corrected.

Line 207-208 plural should be used as multiple genes likely influence lithium response.

Authors’ Response: Corrected.

Lane 208 The reference 41 may be wrong here. there was no information on lithium

Authors’ Response: Corrected. 41 replaced with the following:

Senner F, Kohshour MO, Abdalla S, Papiol S, Schulze TG. The Genetics of Response to and Side Effects of Lithium Treatment in Bipolar Disorder: Future Research Perspectives. Front Pharmacol. 2021 Mar 25;12:638882. doi: 10.3389/fphar.2021.638882. PMID: 33867988; PMCID: PMC8044839.

Line 212 Metastasis is correct?

Authors’ Response: Corrected. By replacing it with cellular homeostasis which means CYP450 play a pivotal role in cellular metabolism and homeostasis. 

Line 221 fatal > fetal

Authors’ Response: Corrected.

Line 221-225 convoluted phrase. please simplify. 

Authors’ Response: In contrast, CYP2C19 is expressed in human fetal brain and disappears after birth, therefore CYP2C19 involves in brain neurodevelopment and significantly influences adult depressive phenotypes. In particular, the absence of CYP2C19 is correlated with a lower prevalence of depression. For example, one the most common CYP2C19 alleles among Swedish subjects is CYP2C19*2. It is characterized by an inactive CYP2C19 enzyme, thus lowering the susceptibility to have depressive mood.

Line 230-232 while interesting as potential biomarker of cyp2d6 metabolic state, i feel the part regarding platelets could be moved on paragraph 2.2 from 2.(as paragraph 2. is mainly focused on  brain).

Authors’ Response: Corrected. The part regarding platelets moved on paragraph 2.2 from 2.

Best Regards

(Corresponding Author)

Round 2

Reviewer 2 Report

With the title change and the minor corrections applied I have no other suggestions to make. I thank the authors for their work.